# The Influence of Relational Benefits on Behavioral Intention and the Moderating Role of Habit: A Study in a Personal Service Business

**DOI:** 10.3390/bs13070565

**Published:** 2023-07-07

**Authors:** Mohammad Karami, Şerife Zihni Eyüpoğlu, Ahmet Ertugan

**Affiliations:** Department of Business Administration, Faculty of Business and Economics, Near East University, 99138 Nicosia, Cyprus; serife.eyupoglu@neu.edu.tr (Ş.Z.E.); ahmet.ertugan@neu.edu.tr (A.E.)

**Keywords:** relationship marketing, relational benefits, habit, revisit intention, personal service business

## Abstract

The intention to repurchase is a key component in relationship marketing. However, minimal attention has been paid to how customers’ habitual behavior moderates the relationship between customers’ evaluation of benefits received from a service provider and the intention to revisit, specifically in a personal service business where customer-service provider interactions likely constitute the core of a sustainable relationship. To address this gap, the current study proposes and tests a comprehensive model to advance the theory of relationship marketing (RM) and additionally contributes to social exchange theory (SET), as well as the theory of repurchase decision making (TRD), in the business service context. Structural equation modeling (SEM) was employed to examine the relationships of the research model. Based on data collected from 482 customers on their perceptions of hairstylists, the empirical findings revealed that relational benefits significantly affect post-experience behavior, satisfaction, trust, and relationship commitment, and subsequently boost the intention to revisit. Furthermore, habit as an unconscious factor moderates the paths between revisiting intention and its determinants. Although several limitations exist, the findings practically and theoretically contribute to the literature on relationship marketing.

## 1. Introduction

In a highly competitive market, it is recommended that relationships with existing customers are maintained and enhanced [1,2]. It has been noted that, for a business firm, it is probable that more profits will be generated by retaining current customers in comparison with attracting and focusing on new customers [3]; it was on this point that the concept of Relationship Marketing (RM) was established, and the strategies for relationship continuum were emphasized for both academicians and marketers [4,5]. The “relationship” is highlighted as the essence of the service business [6]. The service industry has always been relationship-oriented, such that service performance is largely evaluated and developed based on customer–provider relationships and customers’ behavioral intention [6,7,8,9]. Therefore, understanding customer behavior and developing relationship marketing strategies have always been essential in the service industry to retain current customers as a unique value [5,10,11]; this is particularly relevant for the service industry, due to the serious challenges created by the increased competition in the market [2].

Relationship marketing fundamentally aims to create and align all marketing activities with the purpose of building, developing, and maintaining loyal customers for a sustainable relationship [12,13]. To achieve this goal, it is recommended that service firms create/focus on the relational benefits that are likely to be reached through a relationship investment and developed/improved over time [14,15]. Relational benefits positively impact customer satisfaction, trust, and commitment, which in return increases customer intention to repurchase and become loyal [14,16,17,18]. In the developed model of social exchange theory, satisfaction, trust, and commitment are suggested as the significant factors influencing behavioral intention [19,20]. 

Many studies in the service context have been conducted to investigate the impact of purchasing habit on the intention to repurchase and its determinants (e.g., [21,22,23]). For instance, research has reported the moderating effect of habit on relationships between satisfaction and repurchase intention [20,21], trust and repurchase intention [24,25,26], and relationship commitment and intention to repurchase [22]. Although research has indicated that relational benefits directly/indirectly influence customer intention to repurchase [18], according to our knowledge, no research has been carried out to understand the circumstances that might affect the degree (weaker/stronger effect) of behavioral consequences resulting from relational benefits on the intention to repurchase. Do satisfaction, trust, and commitment as the post-experience behaviors of relational benefits influence the repurchase intention of customers with the same effect under different conditions? This study assumes that the relationship between relational benefits’ consequences and customer behavioral intention to repurchase will be dependent on the degree of an unconscious factor such as habit. The proposed assumption of this study is consistent with the theory that habit moderates the determinants of repurchase intention in the service context, as supported in previous studies [22,26]. This approach is also lacking in research on personal services businesses (PSB) such as beauty services, where the frequency and intensity of customer-service provider interactions are high and the quality and continuity of relationships are the core of business sustainability [14,18]. Therefore, the primary goal of this research is to develop and make a contribution to the theories of social exchange, repurchase decision making, and relationship marketing by identifying and testing the proposed relationships by addressing two main questions: First: Does habit play a moderating role in the relationships between repurchase intention and its antecedents originating from relational benefits? Second: How do customer satisfaction, trust, and commitment based on customers’ evaluation of relational benefits impact repurchase intention? Therefore, with a specific focus on personal service businesses, this study empirically investigates the associations between variables in the proposed model based on a sample of 482 valid respondents who were asked about their previous experiences with their service providers.

The rest of this paper is organized as follows. Next part provides a review of the relevant literature, along with definitions of concepts and variables and hypothesis development. In Section 2, the research methodology is described, including sampling, data collection techniques, design of the questionnaire, and measurement of the research variables. The results of analyses such as validity, reliability, and the test of constructs associations are presented and interpreted in Section 3. Section 4 discusses the research findings, theoretical and empirical contribution, and identifies limitations and future research suggestions. 

### 1.1. Literature Review

#### 1.1.1. Relational Benefits

The consumption/usage is the main value of the services or products that customers purchase/receive. However, customers also obtain added value from the relational exchanges, which potentially strengthen the customer-service provider experiences and relationships [15,27]. These additional values, which are formed by loyalty in a long-lasting relationship with the service provider, are defined as relational benefits [16]. The effect of such benefits change over time in a long-term interaction while the customers constantly use the certain product or service [17]. As a result, relational benefits are likely to enhance the existing relationship between service encounters and buyers, which is suggested to enhance relationship sustainability in the long-term relationship [14]. 

Different forms of relational benefits can be received by customers from service providers during relational exchanges, such as social benefits, psychological benefits, economic benefits, and customization benefits [27]. The types of relational benefits have been modified and renamed over time in service studies and categorized into three main dimensions, namely, confidence benefits, social benefits, and special treatment benefits [15]. Confidence benefits are the psychological aspect of the relationships and refer to the benefits that increase the customers’ feeling of security and comfort, reduce their anxiety and uncertainty, and consequently facilitate the mutual understanding between customer and service provider in a relationship [16]. The emotional side of the relationship is explained by social benefits, which develop the friendship between customers and employees over time [28]. Social benefits enhance the feeling of being familiar, individually recognized, and socially supported by the service providers [15,16], specifically in services such as hairdressing where the interpersonal interaction is extremely intense and vital for effective performance [18]. 

Special treatment benefits include price discounts (economic benefits), faster services, and individualized services (customized benefits) that a customer exclusively receives from the serviced provider compared to the other customers [16,29]. These benefits give customers the feeling of being different, better, and valuable in contrast to the other customers, and as a result, make them thankful, grateful, and more satisfied [17]. In the relevant studies on relational benefits, there has been frequent attention on the modified classification of relational benefits’ dimensions (confidence benefits, social benefits, and special treatment) (e.g., [17,18,30,31]). Therefore, this study also uses the same perspective in studying the impact of relational benefits on behavioral intention. 

#### 1.1.2. Post-Experience Behaviors

In the literature, it is stated that post-experience behaviors such as satisfaction (emotional evaluation), trust, and commitment (rational evaluation) are the consequences of customer’s evaluations based on service provider’s behavior in social exchange [19,26], which are recognized as essential aspects of relationship marketing [32,33]. Satisfaction is an important aspect of the buyer–seller relationship, which has become critical for relationship continuity [33,34], particularly in service-oriented businesses [35]. Customer satisfaction is the overall appraisal of customers, which indicates their feelings after experiencing a particular product/service [36,37]. In the service context, satisfaction is measured based on the degree of a customer’s positive feelings about the service provider, which means that it is vital for service providers to constantly monitor the customers’ opinions towards services over time [38]. 

As a related construct, trust has been shown to be a key aspect of relationship marketing [39,40]. Trust is conceptualized as a psychological state that explains the degree of a customer’s confidence in an exchange partner based on the belief that the partner is honest and reliable toward activities that were promised to be delivered [41,42]. Trust in the service context is specifically defined as a customer’s belief in a service provider who performs the tasks appropriately and does not take unexpected actions resulting in negative outcomes [43,44,45]. Trust plays crucial roles in both initiating [46,47] and maintaining the stages of the customer life cycle in the business [17,48,49]. Therefore, it is still necessary to understand the role of trust in the relationship, specifically in the service industry where service provider–customer interaction is important for a sustainable relationship. 

In relationship marketing, commitment has always been an important concept. Commitment refers to a customer’s tendency to continue the formed interaction with an exchange partner [16,41]. Commitment is the voluntary willingness of customers to maintain a relationship in response to the psychological motivation from the organization or service provider [50]. As a result, customers are more willing to invest more in the relationship in the long term [51]. The commitment of a customer is explained by indicators such as the sense of belonging to the service, the feeling of pride to be a customer of a specific service, and also the intention to maintain a boundless relationship to show loyalty [52]. Higher commitment results in the customer not seeking out other organizations that offer similar services [53]. Commitment can be categorized as continuance commitment, normative commitment, and affective commitment [54]. While continuance commitment is described based on customer interest in any financial benefit from an organization, normative commitment is an obligation form of commitment that forces a customer to be committed to the service provider or organization [55]. The customer’s emotional attachment resulting from service provider’s behavior in a relationship is called “affective commitment”. This type of commitment in the context of service provider behavior tightens the customer to the service provider, especially in the long-term relationship [54,56]. Due to this fact and the purpose of this study, affective commitment was chosen to explain relationship commitment in this study [57].

#### 1.1.3. Habitual Behavior 

Habit refers to an unconscious behavior that causes an automatic reaction/response by an individual [58] without any rational pre-assessment to carry out a specific activity [21]. Habit is also postulated as an automatic response learned from previous experiences to the specific situation that an individual performs towards reaching a certain goal [59]. In the context of service, habit is defined as the behavioral tendency resulting from a satisfactory experience of previous purchasing, which leads to buying the same product/service without a conscious mental process [25,60]. Therefore, customer purchasing habit reflects the history of interacting with the service provider and is developed through repeated performance and satisfactory results in post-experience behavior [26,61,62]. According to previous literature, the connection between habit and the intention to repurchase can be explained from three different perspectives; habit exerts a direct effect on the repurchase intention of customers [63,64], mediates the association between repurchase intention and its determinants [65,66], or moderates the repurchase intention and its antecedents [20,26,67]. This study aims to investigate the role of habit as moderator. 

#### 1.1.4. Revisit Intention 

Over time and in the face of competitive market conditions, numerous studies have been carried out to explore, examine, and develop the theory of repurchase decision making (TRD), which demonstrates the importance of this concept in the field of marketing (e.g., [24,68,69,70]). Repurchase intention can be summarized as the optimistic probability that an individual will continue to buy products from the same store or seller in the future after evaluating the experience [25,71]. Repeat purchase intention is when an individual chooses to continue using the same service from the same service provider at the next visit [18,69]. Revisit intention is a similar concept to repurchase intention that indicates the willingness of customers to repeatedly visit the same place, destination, or person due to satisfactory experiences [72]. In the service industry, the term revisit intention is characterized as the degree of a consumer’s desirability to visit the same service provider such as a hairdresser [18], and service centers like restaurants [73] or hotels [74]. Since retaining existing customers is more cost-effective in comparison with attracting new customers, marketing managers and practitioners in the service industry are interested in establishing sustainable relationships focusing on understanding the factors of consumer revisit intention [75].

### 1.2. Hypotheses and Research Model

#### 1.2.1. Relational Benefits and Post-Experience Behaviors

Researchers have demonstrated a strong relationship between relational benefits and customer satisfaction, trust, and relationship commitment in the service context (e.g., [17,18,29,76]). It is argued that, in addition to influencing the customers’ perceptions of service, rational benefits also affect their satisfaction level [77]. Furthermore, the results of conducted research has supported the positive association between the components of relational benefits and customers’ satisfaction [30,78,79], which is also supported by the findings of previous studies [80,81]. As argued in the literature, trust as another post-consumption consequence can also be affected by the relational benefits components in the long-term relationship [17,27]. Using several alternative models, the findings of conducted research have demonstrated that three dimensions of relational benefits positively influence trust in the service industry [78]. In the literature, the association between relational benefit dimensions and relationship commitment has been found to be significantly positive [18,31,82].

More specifically, confidence benefits were shown to be the most influential factor on customer satisfaction [83], which was also concluded in the findings of another study [84]. In the service experience, confidence benefits are suggested to be a generator of customers’ positive sense of security toward what has been promised by the service provider [17,18]. In addition, it has been reported that when the service provider generates high positive feelings, the customers’ willingness to be committed is higher in the relationship [28,31]. According to the previous findings, we hypothesize the following: 

**H1a:** 
*Confidence benefits positively affect customer satisfaction.*


**H1b:** 
*Confidence benefits positively affect customer trust.*


**H1c:** 
*Confidence benefits positively affect relationship commitment.*


Research has also found that the social benefits have significant relationships with customer satisfaction [30,78]. Moreover, social benefits such as being personally recognized and familiarity with the service provider may generate a feeling of trust [85,86], resulting in an increase in the level of trust of customers toward the service provider in the relationship [18,87]. Additionally, it has been shown in previous studies that the willingness to support and maintain relationships among customers who have an excellent relationship with the service provider is higher than those with a normal relationship [88]. Hence, it is suggested that service providers develop relationships with customers to reach the highest level of commitment (e.g., [18,77]). Relevant studies have demonstrated that social benefits are positively associated with the customer’s commitment to the relationship [57]. However, it is interesting that Ko [89] found no significant association between social benefits and relationship commitment. Thus, we hypothesize the following:

**H2a:** 
*Social benefits positively affect customer satisfaction.*


**H2b:** 
*Social benefits positively affect customer trust.*


**H2c:** 
*Social benefits positively affect relationship commitment.*


It was explored whether the special treatment of customers results in customer satisfaction due to the increase in the emotional barrier to switch intention [79,90]. In addition, special treatment benefits were suggested as likely drivers of trust for highly experienced customers in the relationship [17,77]. Also, it was argued that the higher level of special treatments customers receive leads to a higher level of commitment to the relationship with the service provider [91]. Therefore, the special treatments can positively affect the commitment of customers towards the relationship with the service provider [18,57]. Therefore, this study develops the following hypotheses:

**H3a:** 
*Special treatment benefits positively affect customer satisfaction.*


**H3b:** 
*Special treatment benefits positively affect customer trust.*


**H3c:** 
*Special treatment benefits positively affect relationship commitment.*


#### 1.2.2. Post-Experience Behavior and Revisit Intention 

Customer satisfaction has been postulated as a crucial determinant of repurchase decision [92,93]. Customer satisfaction has been shown to be a key predictor of customer loyalty [94] which positively influences the intention to repurchase [95,96,97]. It is also concluded that a strong intention to repurchase is usually possessed by satisfied customers [47,98], while dissatisfied customers are likely to be uncertain about returning to the same place for purchasing and may also switch to another competitor [47,99]. For face-to-face services, the results of conducted studies determined that customer satisfaction is an antecedent for the intention to revisit [100,101]. Therefore, it is hypothesized that: 

**H4:** 
*Satisfaction is positively related to revisit intention.*


The relevant literature has reported the significant impact of trust and commitment on behavioral intention to repurchase [102,103]. For example, in the service context, trust is determined to be a vital factor in increasing the repurchase intention [46,104]. It was shown that the motivation for customers to maintain the relationship declines when the trust and confidence in service providers decrease in the view of service users [105,106]. Other studies on the service industry also empirically support the significant effect of trust on intention to repurchase [107,108,109,110]. In addition to trust, commitment has been determined to be a strong predictor of intention to revisit [18,73]. Affective commitment as a form of commitment was demonstrated to be a factor that significantly affects the intention to repurchase in the service context [56,57,111]. These considerations lead to the following hypotheses: 

**H5:** 
*Trust has a positive influence on intention to revisit.*


**H6:** 
*Relationship commitment has a positive influence on intention to revisit.*


#### 1.2.3. Moderating Effect of Habitual Behavior

The outcomes of previous studies have demonstrated that habit has a moderating role on the effect of trust, satisfaction, and commitment on behavioral intention such as repeat purchase intention [20,21,22,67,112,113]. Empirical evidence supports that the level of habitual behaviors impacts the influence of satisfaction on the intention to repurchase [21]. It was noted that the willingness to repurchase the same service/product among satisfied customers with a high level of habit is higher in comparison to those without such habit. Therefore, satisfaction may not necessarily be a driver of the intention to repurchase from the service place if no shopping habit has been formed. Conversely, the results of some studies indicate that habit weakens the association between satisfaction and behavioral responses [20,67,112,113]. Therefore, 

**H7a:** 
*Habit increases the impact of satisfaction on revisit intention.*


For the trust-repurchase intention relationship, habit has been investigated as a major moderator [20,25]. When a service provider repeatedly performs the same behavior or action and it becomes habitual, the ability of the trust to reduce uncertainty decreases [114], since habit removes the conscious awareness of uncertainty [115]. Therefore, habit in a formed relationship gradually increases and consequently decreases the importance of trust over time [25], which shows that habit negatively impacts the relationship between trust and intention to repurchase [20]. Similar to trust, habit negatively moderates the effect of commitments on the intention to repurchase [25]. The empirical evidence demonstrates that, for customers with a low degree of habit, commitment has an impact on the intention to repurchase [22]. Therefore, we propose the following hypotheses.

**H7b:** 
*Habit reduces the impact of trust on revisit intention.*


**H7c:** 
*Habit reduces the impact of relationship commitment on revisit intention.*


Based on the reviewed literature and hypotheses development, this study proposes a model to graphically present the hypotheses and relationships between variables (Figure 1).

## 2. Methodology

### 2.1. Population and Sample 

This study concentrated on a personal service business (PSB) in the beauty industry. The visitors/customers of the hairdressing salons in Nicosia, Cyprus, formed the population for this study. This city has a population of around 200,000 which is about 16.6% of the country’s total population. The logic behind this choice is explained through three reasons: Firstly, many studies have focused on services provided by hairdressers, which have highlighted the importance of this service in marketing studies (e.g., [17,18,30,68,116,117,118,119,120]). Secondly, the results of previous investigations focused on a group of selected face-to-face services showed that the highest level of interpersonal contact and also the most frequent visits in a customer-service provider relationship occur in the hairdressing services [17,18]. Since both the level of interaction and frequency of visits are at the core of habit and the relational benefits concepts [60], the aim of this study can be more accurately achieved by focusing on the relationship between service providers and customers in the hairdressing services. Thirdly, in recent years, increased competition has been witnessed in beauty services in Cyprus. Statistics have illustrated that, among the European Union members, the highest share of hairdressers and beauticians (2.3% of total employment) was recorded for Cyprus, followed by Malta and Portugal (both 1.3%), and Ireland, Greece, Spain, and Italy (all 1.2%) [121].These figures show the significant trend in terms of investments in beauty services that have continued to increase over the years, which may cause some doubt as to the quality of management in this service sector, especially during and after the pandemic (COVID-19) when beauty services continued to operate, but access to such services was limited due to the imposed restrictions [122].

### 2.2. Data Collection

To test the model, the quantitative data were collected through a self-administrative questionnaire in January and February 2023. The convenience sampling method was used to collect the data. The questionnaire was designed using a Google Forms survey. In the first stage, a pilot test with a sample of 30 participants was conducted to check the validity and reliability of the research instrument. In addition, some professional hairdressers and marketing scholars were asked to review the questionnaire. Based on the pilot study results and experts’ comments, several corrections were made to refine the format, length, rephrasing, readability, and clarity of the instruments. 

With the confidence level of 95% and ±5 margin of error, a sample size of 384 was calculated based on the population of this study. This number was higher than the minimum required sample sized suggested by previous scholars such as Hair et al. [123] (between 150–400), Beavers, et al. [124] (between 150–300), and also Wilson et al. [125] (exceeded than 200). Since the large sample size was suggested for SEM analysis [126] and also to reach a high level of respondent rate, the sample size in this study was expanded to 500. Therefore, in the second stage, a total of 500 questionnaires were shared online to the potential respondents. In return, 482 completed questionnaires were received in a month, which equated to a satisfactory response rate of 98 percent. Using a filtering question, the data of 28 respondents who had changed their hairstylists/beauty salons within the last 12 months were removed and eliminated from the analysis. Therefore, 454 valid questionnaires were used to test the model. 

### 2.3. Questionnaire Construct 

In the first part of the questionnaire, the respondents were asked about their age, gender, education, income, and frequency of visit. The demographic characteristics of the respondents show that a majority (53.1 percent) of the respondents were female. The majority (35.7 percent) of respondents were between 31 and 40 years old. In terms of education, 62 percent of the respondents held a master’s degree or above, 35.9 percent held a bachelor’s, and 2.1 percent education was less than a bachelor’s degree. Furthermore, most of the respondents were single (58.1 percent). More than 94.2 percent of the respondents had not changed their hairstylists or beauty salons within the previous 12 months. In addition, the highest frequency of visits recorded was 9–12 visits per year (39.8 percent). In the second part, the respondents were asked to answer all items related to each construct, according to their experience with the service provider. The demographic data are presented in Table 1.

### 2.4. Measure

The operational definitions of constructs are presented in Table 2. To ensure content validity, all the research constructs in the questionnaire were adapted from the previous studies in the related literature. A slight modification was applied to each item of the constructs to align them with the context of hairdressing services. Each construct was measured based on five-point Likert-type questions (1 reflected “strongly disagree” and 5 reflected “strongly agree”).

The dimensions of relational benefits were measured by items adopted from Vázquez-Carrasco and Foxall [30]. In addition, items employed to measure asset satisfaction were adopted from Jalilvand et al. [127]. Trust was evaluated employing items adapted from De Wulf et al. [33], while relationship commitment was evaluated utilizing items adapted from Dagger and O’Brien [17] and Chou and Chen [18]. Habit was measured by items adapted from Chiu et al. [25]. Finally, items adapted from Chen et al. [101] were used to evaluate revisit intention. Constructs and related items in the questionnaire are presented in Table 3.

### 2.5. Data Analysis 

To analyze the data and also examine the associations between constructs in the proposed model, the statistical tools of SPSS (version 24) and AMOS (version 24) were used. Goodness of fit was checked given the values of the model-fit measures. The fit indices and acceptable values recommended in prior studies are shown in Table 4.

### 2.6. Common Method Bias

When all constructs (both independent and independent variables) are measured based on the data collected from a single survey, common method variance (CMV) can affect relationships [131]. Therefore, as a prior step before assessing the reliability and validity, the examination of bias in the data was advised [132]. Therefore, Herman’s single factor test, a broadly accepted test for assessing common method bias (CMB), was used to check whether a single factor could account for the majority of the variance [131]. The analysis was run using all constructs, and the results of the test revealed that 39.13% of variance was accounted by the first factor, which is less than the maximum accepted limit of 50% suggested by Herman [131,133]. Thus, it can be conclude the data are free of any common method bias. 

## 3. Results

### 3.1. Tests of the Measurement Model

To test whether the constructs and items are well fit with the hypothesized model, measures of fit were tested based on collected data using Confirmatory Factor Analysis (CFA). The outcome indices of χ^2^/df = 2.575, GFI = 0.96, AGFI = 0.92, NFI = 0.95, CFI = 0.98, and RMSEA = 0.036 exceed the recommended thresholds, which indicates an adequate fit to the collected data. 

The constructs’ reliability and validity were examined using internal consistency, convergent validity, and discriminant validity analysis. Therefore, the values of Cronbach’s α, composite reliability (CR), and average variance extracted (AVE) were checked for this purpose. Firstly, for reliability analysis in order to assess the model’s internal consistency, the tests of Cronbach’s α and composite reliability (CR) were conducted. According to the results of the Cronbach’s alpha and CR tests, all values exceeded the acceptable limit of 0.07, as suggested in previous studies [134,135]. Therefore, the internal consistency of all constructs was adequate, which confirmed the desirable reliability of the research. The values of Cronbach’s α and composite reliability (CR) are reported in Table 5.

Secondly, it was recommended in prior studies that, to ensure the convergent validity, the values of factor loadings should be higher than 0.5 for all items, and, for all constructs, the CR and AVE values should be higher than 0.7 and 0.5, respectively [127,130]. Given the outcomes reported in Table 5, all indicators for factor loading, CR, and AVE in the measurement model exceeded the acceptable thresholds, which ensured the adequate condition for convergent validity.

Finally, the pairwise construct comparison method was performed to check the discriminant validity [134,136]. This method presents a matrix of square roots of the AVE and correlation coefficients between constructs for comparing the values. To ensure discriminant validity, diagonal indices in the matrix (the square roots of the AVE for each construct) should be higher than the off-diagonal indices (values of correlations coefficients of a construct to other constructs). The indicator of all correlations between constructs in addition to the square roots of the AVE for each construct demonstrated that the conditions for discriminant validity were satisfactorily met (Table 6). Furthermore, the value of Heterotrait–Monotrait ratio (HTMT) was checked to confirm/support the discriminant validity. This method was used due to its advantages over the prior studies in a variety of conditions [71,137]. According to the result, the values of HTMT were less than the acceptable limit of 0.9, confirming no discriminant validity [135] (Table 6). Therefore, satisfactory reliability, convergent validity, and discriminant validity of the measurement model used in this study were confirmed. 

### 3.2. Tests of the Structural Model

As a powerful valid multivariate technique, structural equation modeling (SEM) was employed to test the hypotheses in the proposed model (K. Y. Lin et al., 2017). The indices for the fit of the structural model (χ2/df = 2.24, GFI = 0.91, AGFI = 0.88, NFI = 0.90, CFI = 0.94, and RMSEA = 0.039) showed an acceptable model fit. The path coefficients of the respective constructs, the levels of significance, and variances (R^2^) are summarized in Figure 2.

The results demonstrated that all hypotheses were significant, as the *p* value was higher than 0.05. The H1a–H1c supported the significant positive relationship between confidence benefits and satisfaction, trust, and relational commitment (β = 0.35, *p* < 0.001; β = 0.41, *p* < 0.001; β = 0.37, *p* < 0.001). Moreover, the relationship between social benefits and satisfaction, trust, and relational commitment are all supported by H2a: (β = 0.26, *p* < 0.05), H2b: (β = 0.18, *p* < 0.10), and H2c: (β = 0.16, *p* < 0.10). Hence, the special treatment benefits are positively related to satisfaction, trust, and relational commitment as H3a, H3b, and H3c are all supported (H3a: β = 0.39, t < 0.001; H3b: β = 0.25, *p* < 0.05; H3c: β = 0.29, *p* < 0.05). Furthermore, the relationship between customer satisfaction, trust and commitment, and revisit intention are supported by H4 (β = 0.38, *p* < 0.001), H5 (β = 0.44, *p* < 0.001), and H6 (β = 0.24, *p* < 0.05). The explained variance (R^2^) of revisit intention was 45 percent. For satisfaction, trust, and commitment, the variances were 0.36, 0.54, and 0.52 percent, respectively. All of the variances indicate a medium effect size of R2, since the indices exceed the cutoff value of 0.13 [137]. 

### 3.3. Test of the Moderating Effects

The examination of the moderating effect of habit was conducted using the sub-group analysis method following Wang et al. [138]. Therefore, the respondents were categorized into two “low-habit” (*n =* 218) and “high-habit” (*n =* 236) subgroups based on their responses to each item of habit in the survey form. For the purpose of grouping, K-means cluster analysis was used, as the sample size was larger than 200 respondents in this study [123]. Firstly, multi-group analysis was utilized to examine whether any differences exist between groups. The results of the test for two constrained and unconstrained models indicated a significant difference between low-habit and high-habit subgroups at the significant level of *p* < 0.001. Secondly, to test the hypotheses, the invariance levels of paths were checked to determine the difference in the path coefficients between the low-habit subgroup and high-habit subgroup, as proposed by Yoo [139]. Therefore, the test of X^2^ differences was performed as an indicator to check the equality of the path coefficients in the base model compared to the other two models (Model 1: base model; Model 2: model including the low-habit group; Model 3: model including the high-habit group). The indices of model fit indicated a satisfactory level of fit in all three models. The results of the test provided in Table 7 show a significant difference of X^2^ in model 2 (Δx^2^a = 13.307) and also model 3 (Δx^2^b = 7.394) to the base model, both at the significance level of *p* < 0.05. The results indicated that habit moderated the path between satisfaction, trust, and relationship commitment to revisit intention. Therefore, H7a–H7b are supported.

As presented in Figure 3, the path coefficient indicators for the high-habit subgroup (path of satisfaction to revisit intention: β = 0.09; *p* > 0.001; trust to revisit intention: β = 0.17 *p* < 0.05; relationship commitment to revisit intention: β = 0.11; *p* > 0.05) are lower than those for the low-habit subgroup (path between satisfaction and revisit intention: β = 0.21 *p* < 0.10; trust and revisit intention: β = 0.27 *p* < 0.05; relational commitment and revisit intention: β = 0.17 *p* > 0.05).

## 4. Discussion and Conclusions

### 4.1. Findings of the Study

This study proposed and empirically examined a comprehensive model to understand how habit as an unconscious factor influences the relationship between customers and the service provider in the personal service context in a long-lasting relationship. Therefore, a test was first conducted to confirm the relationship between the relational benefits dimensions and post-experience behavior such as satisfaction, trust, and commitment, and subsequently, their impact as determinants of revisit intention. Secondly, the role of habit as a moderator on the path between revisit intention and its determinants was investigated. Overall, the results revealed a positive sign between all relationships as well as the moderating role of habit in the proposed model.

In detail, the results show that confidence benefits as a dimension of relational benefit was a prior factor affecting satisfaction, trust, and commitment. These results are consistent with the outcomes of previous studies in the literature (e.g., [16,57,83]). Furthermore, treatment benefits significantly impact satisfaction, trust, and commitment respectively, which was previously confirmed in the relevant studies (e.g., [16,57,77]). Additionally, with a lower significance level, the findings indicated that social benefits have a significant positive relationship with satisfaction, trust, and commitment; these results support the outcomes of previous studies (e.g., [57,78,86]).

All these results are evidence of the important role of relational benefit dimensions in the personal service context in terms of explaining the customer-service provider relationship. This study determined that satisfaction, trust, and commitment are crucial factors that have a significant positive effect on revisit intention. It was also confirmed in the results of previous studies in the literature that if the level of satisfaction, trust, and commitment increases, the intention to revisit is also increased [16,18,29,76].

Most importantly, the results determined that habit plays a moderating role in the relationship of satisfaction, trust, and relationship commitment with customers’ intention to revisit. Firstly, with regard to satisfaction, the results indicated that the moderating influence for the subgroup with high habit was greater than the low-habit subgroup, which supports the theory that satisfaction exerts a lower impact on intention to revisit for customers with a high level of habit (e.g., [20,67]). However, this finding contradicts the outcomes of some previous studies (e.g., [21]). Therefore, this study confirmed that customers with a high level of habit are not as sensitive to satisfaction in the relationship with their service provider for their future behavioral intentions compared to lower-habit customers [23].

Secondly, the results showed that habit greatly affected the path between trust and customers’ intention to revisit the subgroup of low-habit customers. The findings indicated a significant moderating role of habit for the customers with a high level of habit in the trust-revisit intention association [20]. Therefore, this finding suggests that an unconscious factor like habit might gradually dominate the effect of trust in the long-term relationship. In return, habitual behavior might negatively affect the customers’ perception of trust and consequently decrease the importance of trust in the future behavioral intention in the interaction with the service provider [25].

Finally, according to the findings, habit highly influences the path between commitment and revisit intention for customers with habit compared with high-habit ones, thus confirming the strong moderating role of habit for customers with a high habit in the relationship between commitment and revisit intention [22]. It can be suggested that habit, as an unconscious factor, gradually overcomes the role of commitment as a strengthening factor in the relationship over time. In return, the habit will act as a crucial factor to maintain and enhance the relationship.

### 4.2. Theoretical Implications

This study theoretically contributes to the literature in various ways. The significant role of relational benefit dimensions in the customer-service provider relationship context has been examined in previous studies (e.g., [16,30,57]). Based on the author’s knowledge, few studies have been conducted by combining these dimensions with satisfaction, trust, and commitment with the purpose of simultaneously examining the factors that influence customers’ intention to revisit and studying the moderating role of habitual behavior in the social exchange context, specifically in the personal service context where interactions play an essential role in the quality and form of the relationship between the customer and service provider. Therefore, the current study contributes to the social exchange theory and advances relationship marketing from two perspectives. First, it provides evidence of the important role of relational benefits as the key driver in a sustainable relationship in long-term interactions. Second, this study contributes to relationship marketing theory by exploring the power of a customer’s unconscious mind in a relationship, focusing on habitual behavior as a key driver of relationship maintenance [20,25,67]. Therefore, by developing a research model, habit has been considered as a moderator between the paths of revisit intention and its determinants and can be considered as another crucial parameter in relationship marketing theory. The results expand relationship marketing theory given the evidence provided that habit strongly maintains the relationship between customers and their service providers, which was formed by relational benefits and developed by satisfaction, trust, and commitment.

### 4.3. Managerial Implications

The current study was conducted to contribute to the marketing relationship context by studying consumers’ intentions to revisit the service providers in the personal service businesses, especially in a hairstyling service where face-to-face interaction essentially forms, builds, and maintains the relationship. The findings of the present study provide some important empirical implications for managers, marketers, academicians, and also practitioners in the relevant context.

The marketers and practitioners in the personal service industry need to develop each dimension of relational benefits accordingly to reach a desirable and satisfactory level of satisfaction, trust, and commitment among customers, specifically by examining confidence benefits, special treatment benefits, and social benefits as prior factors affecting satisfaction, trust, and commitment, respectively. Therefore, confidence benefits should be given more attention in delivery services, which is likely to be fostered by increasing the feeling of security or reducing anxiety to make customers more satisfied, giving them a feeling of trust, and making them more committed to continue their relationship with their service providers. As a consequence, the customers are more likely to repeat the behavior intention such as revisiting the same place. Second, an increase in the special treatment benefits should also be achieved. Individually offering more discounts on prices, delivering fast services, or customized services give the customer a sense of being unique, which ultimately positively influences the satisfactory level of service, trust, and potential commitment to the service. Eventually, the intention to revisit will become automatically accelerated. Lastly, social benefits, as another influential factor determining post-experience behavior, should receive greater attention from managers and marketers. An increase in familiarity, frequent personal recognition, and also strengthening a friendly relationship with customers increase the potential of the service provider to penetrate the customer’s decision by giving them the feeling of confidence. In return, the customer’s feeling of satisfaction, trust, and commitment towards the service/service provider possibly increases such that the customer’s revisit intention is enhanced.

Most importantly, this study highlighted the role of habit as a crucial factor in maintaining the relationship between customers and service providers in long-term interactions. The findings of this study confirmed that an unconscious mind possibly dominates the cognitive and effective behavioral processes/outcomes such as satisfaction, trust, and commitment, especially when a relationship is formed and continues based on desirable experiences. The results show that the influence of satisfaction, trust, and commitment decreases when the habitual behaviors increase in the long term. Therefore, service providers should encourage customers to revisit the service place automatically when the post-experienced behavior is built and reaches a sufficient level, as suggested in prior research [25]. The service providers could encourage customers to make more frequent visits by providing incentive options in addition to the services in order to offer attractive benefits (socially, specially, and confidently). A relationship formed by relational benefits and enhanced by a satisfactory level of experiences can be fostered and maintained by habit. Hence, it is suggested that providers and managers build and boost a strong bond with loyal customers to develop customers’ habit focusing on the dimension of relational benefits.

### 4.4. Limitations and Future Research Suggestion

Despite the important contributions of this study, some limitations exist and are recommended to be addressed in future studies. Firstly, this study was conducted based on data collected from the capital city of Cyprus, where the number of available beauty salons is limited compared to big cities. Therefore, due to the lack of preferences for dissatisfied customers to switch from their current service place/providers to another one, the reliability and validity of the results might be decreased by this fact. Hence, it is suggested that future studies target highly competitive marketplaces where there is a greater availability and accessibility of similar service centers. Additionally, focusing on large cities would allow researchers the opportunity to have a sample size with a diverse background, psychologically and culturally, which may present different values for the antecedents of behavioral intentions. Secondly, this study filtered the sample size to customers who used the service for a specific duration (12 months). Future scholars can expand the duration of service usage to longitudinally examine the impact of the relationship duration on behavioral intention. Thirdly, this study employed a convenience sample. Therefore, it is suggested that authors collect the data using probability sampling in future studies in the same context. Fourthly, adopting more items from different resources, and modifying and testing the reliability and validity of constructs are recommended in future efforts. Lastly, in future studies, researchers can focus on a single or group of other personal services businesses such as clothes shops, travel agencies, fast food chains, cinemas, airlines, banking services, etc. [27].

### 4.5. Conclusions

This study was conducted to develop and examine a comprehensive model explaining the determinants of repeat visit intention in the personal service business. By identifying relational benefits, satisfaction, trust, commitment, and habit, this study tested their impacts on revisit intention. The results revealed that all the dimensions of relational benefits positively affected satisfaction, trust, and commitment, which in return significantly influenced revisit intention. Additionally, the moderator effect of habit on the path between revisit intention and its antecedents was positively confirmed. The findings revealed several implications both theoretically and practically for academicians and practitioners; however, similar to other works, this study also faces several limitations.

## Figures and Tables

**Figure 1 behavsci-13-00565-f001:**
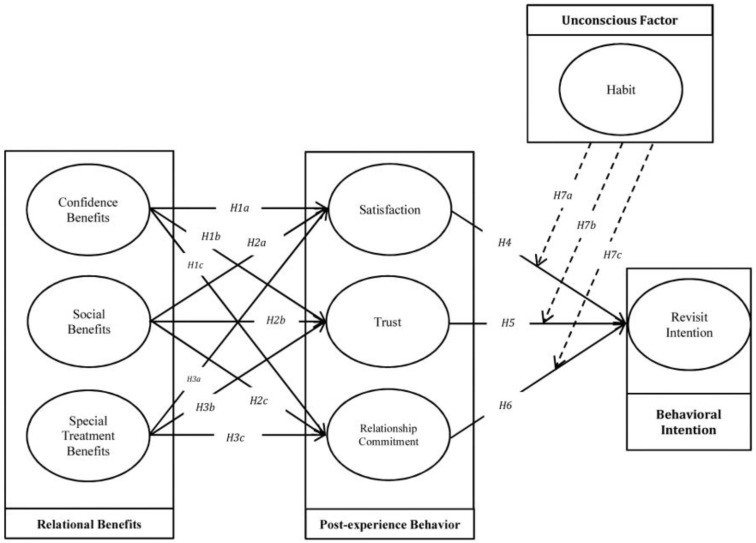
Research Model.

**Figure 2 behavsci-13-00565-f002:**
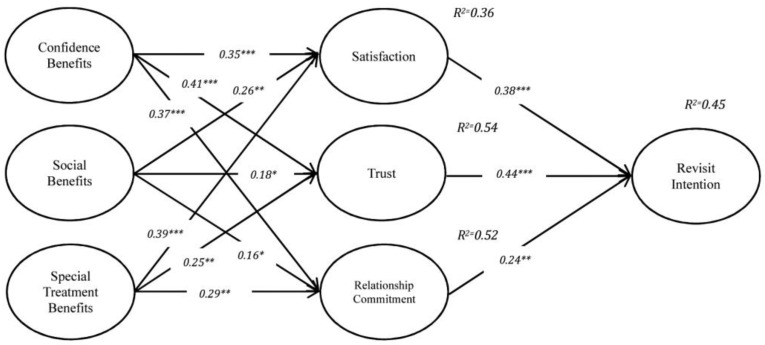
Result of relationships. Notes: * *p* < 0.10; ** *p* < 0.05; *** *p* < 0.001.

**Figure 3 behavsci-13-00565-f003:**
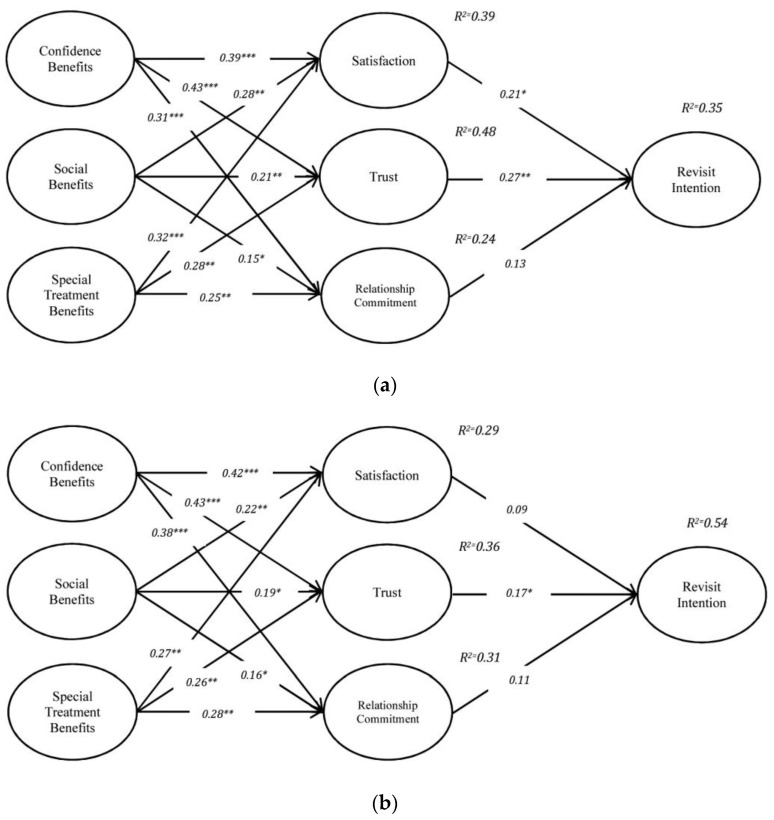
The result of structural model with subgroups of habit. (**a**) Result of low-habit subgroup. (**b**) Result of high-habit subgroup. Notes: * *p* < 0.10; ** *p* < 0.05; *** *p* < 0.001.

**Table 1 behavsci-13-00565-t001:** Demographic profile.

Measure	Item	Frequency (N = 482)	Percentage (%)
Gender	male	226	46.9
	female	256	53. 1
Age	below 21	24	24.0
	21–30	170	37.3
	31–40	182	35.7
	41–50	78	16.2
	51–60	28	5.8
	61–above	0	0
Education	less than bachelor’s	10	2.1
	bachelor’s	173	35.9
	master’s and above	299	62.0
Marital status	single	280	58.1
	married	202	41.9
Monthly income (Euro)	100 or less	10	2.1
	100–200	48	10.0
	200–300	210	43.6
	300–400	154	32.0
	400–500	24	5.0
	500 or more	35	7.5
Frequency of visit/year	less than 3 times	2	0.4
	between 3–6 times	20	4.1
	between 6–9 times	110	22.8
	between 9–12 times	192	39.8
	more than 12 times	158	32.8
Switched within the last year	yes	28	5.8
	no	454	94.2

**Table 2 behavsci-13-00565-t002:** Operationalized constructs.

Construct	Definition
Relational Benefits	The degree of benefit that a customer receives from the relationship with the service provider.
Satisfaction	The degree of an individual’s positive feeling toward the service provider as a result of relational benefits.
Trust	The degree of an individual’s feeling of reliability derived from the benefits received from the relationship with his/her hairstylist
Relationship commitment	The degree of an individual’s feeling of pride and the importance of his/her relationship with her/his hairstylist.
Habit	Habit refers to the degree to which an individual visits the service provider automatically without thinking.
Revisit intention	The degree of an individual’s willingness to visit the service provider in the future.

**Table 3 behavsci-13-00565-t003:** Constructs and items of the questionnaire.

Construct	Measurement Statements	References
Confidence Benefits	1. I believe there is less risk that something will go wrong in this service provider’s performance.	[30]
	2. I feel I can trust this service provider(s).	
	3. I have more confidence that the service will be performed correctly by this service provider(s).	
	4. When I receive the service from this service provider(s), I have less anxiety.	
	5. I know what to expect when visiting this service provider(s).	
	6. I receive the highest level of service from this service provider(s).	
Social Benefits	1. This service provider(s) recognizes me well.	[30]
	2. I know this service provider(s) well.	
	3. I have developed a friendship with this service provider(s)	
	4. This service provider(s) remembers my name.	
	5. I enjoy the social aspects of the relationship with this service provider(s).	
Special Treatment Benefits	1. I receive discounts from this service provider(s) that most customers do not receive.	[30]
	2. I am offered services with better prices by this provider(s).	
	3. I receive special services from this service provider(s) that most customers do not receive.	
	4. This service provider(s) prioritizes my name in the appointments list.	
	5. I receive faster service than most customers.	
Satisfaction	1. My experience this service provider(s) is excellent.	[127]
	2. This service provider(s) always meets my expectations	
	3. Overall, I am satisfied with this service provider(s)	
Trust	1. This service provider(s) gives me a feeling of trust.	[33]
	2. I have trust in this service provider(s).	
	3. This service provider(s) gives me a trustworthy impression.	
Relationship Commitment	1. I really care about my relationship with this service provider.	[16,18]
	2. My relationship with this service provider(s) is very important to me.	
	3. I really commit to my relationship with this service provider(s).	
Habit	1. Vising this service provider(s) has become a routine for me	[25]
	2. Visiting this service provider(s) is something I do without thinking.	
	3. It makes me feel weird if I do not visit this service provider(s) in the future.	
	4. I have been visiting this service provider(s) for a long time.	
Revisit Intention	1. I would revisit this hairdresser again in the near future.	[101]
	2. I am interested in revisiting this hairdresser again.	
	3. I will come back again.	
	4. There is a likelihood that I will revisit in the future.	

**Table 4 behavsci-13-00565-t004:** Fit indices for the measurement and structural models.

Fit Indices	Acceptable Value	References
χ^2^/degree of freedom	≤3	[128]
GFI (goodness-of-fit index)	≥0.8	[129]
AGFI (adjusted goodness-of-fit index)	≥0.8	[129]
NFI (normed fit index)	≥0.8	[123]
CFI (comparative fit index)	≥0.9	[130]
RMSEA (root mean square error of approximation)	≤0.08	[130]

**Table 5 behavsci-13-00565-t005:** The indicators of reliability and validity.

Construct	Item	Mean	Factor Loading	AVE	CR	Cronbach’s α
Confidence Benefits (CB)	CB1	4.13	0.89	0.70	0.93	0.95
	CB2	4.29	0.76			
	CB3	4.15	0.95			
	CB4	4.16	0.93			
	CB5	4.23	0.79			
	CB6	4.33	0.68			
Social Benefits (SB)	SB1	4.38	0.71	0.60	0.88	0.87
	SB2	4.49	0.79			
	SB3	3.43	0.90			
	SB4	4.83	0.69			
	SB5	4.37	0.77			
Special Treatment Benefits (STB)	STB1	4.08	0.71	0.58	0.87	0.94
	STB2	4.04	0.79			
	STB3	3.78	0.90			
	STB4	3.59	0.63			
	STB5	3.83	0.77			
Satisfaction (SA)	SA1	4.36	0.93	0.75	0.89	0.91
	SA2	4.36	0.95			
	SA3	4.51	0.70			
Trust (TR)	TR1	4.46	0.90	0.79	0.91	0.93
	TR2	4.68	0.87			
	TR3	4.67	0.90			
Relationship Commitment (RC)	RC1	4.53	0.91	0.78	0.91	0.96
	RC2	4.50	0.91			
	RC3	4.47	0.84			
Habit (HA)	HA1	4.20	0.90	0.82	0.94	0.97
	HA2	4.21	0.98			
	HA3	4.10	0.82			
	HA4	4.24	0.92			
Revisit Intention (RI)	RI1	4.26	0.91	0.80	0.94	0.91
	RI2	4.26	0.89			
	RI3	4.27	0.82			
	RI4.	4.37	0.97			

**Table 6 behavsci-13-00565-t006:** The matrix of pairwise construct comparison.

Construct	CB	SB	STB	SA	TR	RC	HA	RI
CB	**0.836**							
SB	0.536	**0.774**						
STB	0.554	0.629	**0.761**					
SA	0.646	0.477	0.656	**0.866**				
TR	0.451	0.552	0.592	0.479	**0.888**			
RC	0.521	0.654	0.543	0.516	0.580	**0.883**		
HA	0.592	0.357	0.626	0.574	0.513	0.402	**0.905**	
RI	0.708	0.451	0.408	0.618	0.417	0.488	0.782	**0.894**

Notes: Bolded values represent the squared root of AVE. Below the diagonal are correlation coefficients; level of significance is *p* < 0.05.

**Table 7 behavsci-13-00565-t007:** Invariance Level.

		X^2^	df	X2/df	GFI	AGFI	NFI	CFI	RMSEA
Model 1	Base model	1224.905	476	2.573	0.89	0.88	0.78	0.92	0.038
Model 2	Effect of Low-habit	1238.212	475	2.606	0. 88	0.87	0.77	0.86	0.037
Model 3	Effect of High-habit	1232.299	475	2.594	0.88	0.87	0.77	0.92	0.037
Δx^2^ a: x^2^ model 2 − x^2^ model 1 = 13.307			Δdf a: df model 2 − df model1 = 1				*p* < 0.05		
Δx^2^ b: x^2^ model 3 − x^2^ model 1 = 7.394			Δdf b: df model 3 − df model1 = 1				*p* < 0.05		

## Data Availability

Data will be made available upon from the corresponding authors upon reasonable request.

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
