# Peer review of "The Influence of Relational Benefits on Behavioral Intention and the Moderating Role of Habit: A Study in a Personal Service Business"

_behavsci, 2023, doi:10.3390/bs13070565_

Round 1

Reviewer 1 Report

See PDF.

Author Response

"Please the attachment"

Reviewer 2 Report

1) Please provide additional information about:

- the reason for choosing the sample size of 500

- on which bases the segmentation variables of the sample from Table 1 were chosen?

- are the segmentation variables of the sample size from Table 1 statistically significant according to the population?

 2) It is not clear whether the sample size is statistically significant for the population or not.

1) Please, revise the writing of:

- “revisit”. The correct form is “revisiting” (Line 25)

- “make a contribution”. The correct form is “contribute” (Line 26)

- “relationship oriented”. The correct form is “relationship-oriented” ((Line 39)

- “the service”. The correct form is “service” (Line 38)

- “particular”. The correct form is “particularly” (Line 43)

- “increase”. The correct form is “increases” (Line 51)

- “the relationships”. The correct form is “relationships” (Lines 57, 235 and 237)

- “relevant”. The correct form is “the relevant” (Line 84)

- “valued”. The correct form is “value” (Line 95)

- “formed”. The correct form is “are formed” (Line 97)

- “As result”. The correct form is “As a result” (Line 100)

- “encounter”. The correct form is “encounters” (Line 101)

- “service oriented”. The correct form is “service-oriented” (Line 132)

- “towards maintaining”. The correct form is “to maintain” (Line 151)

- “feeling”. The correct form is “the feeling” (Line 155)

- “An emotional attachment of a customer’s resulting from service provider’s behavior in a relationship”. The correct form is “The customer’s emotional attachment resulting from service provider’s behavior in a relationship” (Lines 162-163)

- “with regard to”. The correct form is “concerning” (Line 167)

- “post experience”. The correct form is “post-experience” (Lines 179 and 513)

- “The aim of this study is”. The correct form is “The study aims” (Line 184)

- “probability”. The correct form is “the probability” (Line 191)

- “visit a same”. The correct form is “visit the same” (Line 195)

- “center”. The correct form is “centers” (Line 198)

- “restaurant”. The correct form is “restaurants” (Line 198)

- “hotel”. The correct form is “hotels” (Line 198)

- “cost effective”. The correct form is “cost-effective” (Line 199)

- “Due to the fact that”. The correct form is “Since on” (Line 199)

- “the strong”. The correct form is “a strong” (Line 205)

- “have supported”. The correct form is “has supported” (Line 209)

- “the feeling”. The correct form is “a feeling” (Line 232)

- “intention”. The correct form is “rhe intention” (Lines 268 and 284)

- “level”. The correct form is “level” (Line 289)

- “First”. The correct form is “Firstly” (Lines 324, 422, 534, and 624)

- “Second”. The correct form is “Secondly” (Lines 326 and 542)

- “Third”. The correct form is “Thirdly” (Line 332)

- “A questionnaire”. The correct form is “The questionnaire” (Line 345)

- “habit”. The correct form is “the habit” (Lines 300 and 555)

- “trust”. The correct form is “the trust” (Line 302)

- “On the basis of”. The correct form is “Based on” (Lines 314 and 349)

- “the marketing”. The correct form is “marketing” (Line 328)

- “visit”. The correct form is “visits” (Lines 329 and 366)

- “In  order  to  ”. The correct form is “To” (Lines 343 and 347)

- “per cent”. The correct form is “percent” (Lines 360-366)

- “education”. The correct form is “of education” (Line 363)

- “previous”. The correct form is “the previous” (Line 388)

- “importance”. The correct form is “the importance” (Line 393)

- “this this service”. The correct form is “this service” (Line 403)

- “hairdresser’s”. The correct form is “hairdressers” (Line 403)

- “related with satisfaction”. The correct form is “related to satisfaction” (Line 466)

- “fallowing”. The correct form is “following” (Line 477)

- “was”. The correct form is “were” (Line 481)

- “model to understand”. The correct form is “model for understand” (Line 509)

- “was firstly conducted”. The correct form is “was first conducted” (Line 512)

- “All this results”. The correct form is “All these results” (Line 526)

- “First”. The correct form is “Firstly” (Line 534)

- “to revisit in the subgroup”. The correct form is “to revisit the subgroup” (Line 543)

- “customer”. The correct form is “the customer” (Line 566)

- “of customer’s”. The correct form is “of a customer’s” (Line 570)

- “sufficient”. The correct form is “a sufficient” (Line 614)

- “can possibly be”. The correct form is “can be” (Line 614)

- “with the loyal customers”. The correct form is “with loyal customers” (Line 620)

- “In spite of”. The correct form is “Despite” (Line 623)

- “market places”. The correct form is “marketplaces” (Line 629)

- “is greater”. The correct form is “is a greater” (Line 629)

- “give researchers”. The correct form is “allow researchers” (Line 631)

2) The comma is missing before “and” in the enumerations (Lines 47, 51, 53, 64, 75, etc,)

Reviewer 3 Report

I find the paper topic interesting, proving that the authors know deeply the research topic and used an appropriate methodologies in order to cover the proposed objective of the research.

The paper is well structured and results  clearly presented, including also the limits and future development.

Reviewer 4 Report

1. The "intention to repurchase" is discussed in the hypothesis. However, Literature Review, Model and Scale are "revisit". Totally inconsistent.

2. Please measure non-response bias and common method bias

3. Items lower than 0.7 in Factor Loading cannot be used.

4. The measurement of discriminant validity also needs to provide the value of HTMT

5. Figure 2, R>1?

6. For group adjustment, please use multi-group analysis and compare the differences.

Round 2

Reviewer 4 Report

thank you for respones